# “I Will Not Leave My Body Here”: Migrant Farmworkers’ Health and Safety Amidst a Climate of Coercion

**DOI:** 10.3390/ijerph16152643

**Published:** 2019-07-24

**Authors:** C. Susana Caxaj, Amy Cohen

**Affiliations:** 1School of Nursing, University of Western Ontario London, ON, N6A 5B9, Canada; 2Anthropology, Okanagan College, Vernon, BC, V1B 2N5, Canada

**Keywords:** temporary migrant agricultural workers, farm workers, Seasonal Agricultural Worker Program, occupational health and safety, Temporary Foreign Worker Program, coercion, Canada

## Abstract

Every year more temporary migrant workers come to Canada to fill labour shortages in the agricultural sector. While research has examined the ways that these workers are made vulnerable and exploitable due to their temporary statuses, less has focused on the subjective experiences of migrant agricultural workers in regards their workplace health and safety. We conducted interviews and focus groups with migrant workers in the interior of British Columbia, Canada and used a narrative line of inquiry to highlight two main themes that illustrate the implicit and complex mechanisms that can structure migrant agricultural workers’ workplace climate, and ultimately, endanger their health and safety. The two themes we elaborate are (1) authorities that silence; and (2) “I will not leave my body here.” We discuss the implications of each theme, ultimately arguing that a number of complex political and economic forces create a climate of coercion in which workers feel compelled to choose between their health and safety and tenuous economic security.

## 1. Introduction

Temporary migrant agricultural (MA) workers are a driving force addressing labour shortages in the Canadian agricultural system. In fact, 69,705 individuals [1], roughly 75% of agricultural positions [2], are currently filled by a MA worker. As shortages in the sector continue to increase, temporary MA workers represent a growing group of workers in Canada without the same legal rights and protections as Canadian workers [3]. Many return for several seasons, a large group even remaining MA workers for decades [4]. With no clear pathway for permanent residence, some describe these workers as circular rather than temporary migrants. Yet these individuals remain permanently marginal and vulnerable in their communities, even when they return year after year [5]. Our research focuses on better understanding the workplace environments of migrant agricultural workers in the Okanagan Valley, a key destination for migrant agricultural workers in the province of British Columbia. In particular, we examine how these workplace environments shape migrant agricultural workers’ health.

MA workers typically enter the Canadian job market under two different streams: the Seasonal Agricultural Workers Program (SAWP), or the Temporary Foreign Worker Program Agricultural Stream (TFWP Ag Stream). The SAWP represents various bilateral agreements between the Canadian federal government and the ‘sending’ countries (Mexico and 11 countries from the Caribbean) [6]. Under the SAWP, worker recruitment is overseen by ministries of labour in sending countries, workers must return by December 15th of each year, and can stay for no longer than eight months in Canada [7]. Recruitment under the TFWP Ag is carried out by non-governmental agencies who may charge large fees from hopeful migrants; workers may come from a variety of countries (e.g., Guatemala, India) and may stay for a maximum of two years in Canada [8]. Given that the vast majority of migrant agricultural workers come under the SAWP, our research focuses on workers coming under this program.

In 2018, 16,890 temporary migrant agricultural positions were approved in the province of British Columbia, including placements under the SAWP and TFWP Ag Stream. The Okanagan region hosts the second largest numbers of SAWP workers in the province of BC second only to the Fraser valley. Every year, several thousand workers are placed in this region, representing 80% of Jamaican workers and 30% of Mexican workers in the province [9]. Despite the influx of workers and other immigrants and refugees, the Okanagan continues to be characterized as a White, socially conservative region and, in comparison to larger urban centres, has limited capacity to support the needs of these diverse diasporas [10,11]. This political climate complicates MA workers’ abilities to assert their basic rights in the workplace.

Migrant farm workers receive very little access to protections they are entitled to, even when using public service systems, including health care facilities. Yet farm work presents unique risks to both physical and psychological health. Complaint-driven and pre-arranged inspections have historically been the norm with regulators (i.e., Worksafe, Employment Standards) in the region, a system that puts the onus on migrant agricultural workers to identify a labour or health violation and come forward as a ‘whistle-blower’, an action that can jeopardize their job status. With little to no separation between their workplace and living quarters, common substandard housing conditions such as overcrowding, inadequate washroom and cooking facilities are often accepted as a necessary (albeit unpleasant) component of employment. These environments ultimately threaten the health and safety of migrant agricultural workers.

In this article, we discuss two over-arching themes discussed by migrant agricultural workers that help illustrate the implicit and complex mechanisms that can determine their workplace, and ultimately, endanger their health and safety. Using a narrative line of inquiry, we developed two themes. Theme 1, authorities that silence, encompassed two storied experiences, or sub-themes: “it doesn’t make sense to say anything to them” and “no-one to enforce the rules.” In this theme, active and passive coercion mitigated migrant workers’ ability to ensure a safe and healthy workplace. In Theme 2, “I will not leave my body here,” migrant agricultural workers shared ways in which their bodies were both their key asset and liability in maintaining their livelihoods in agriculture, by, for example, accepting occupational risks as a condition of keeping their employment. These findings help to illustrate how workplace conditions position workers to either accept (e.g., by normalizing hazards), negotiate (e.g., by conceding certain rights in order to maintain employment) and/or resist (e.g., by speaking out or reporting) exploitative or unsafe norms and practices. These themes also illustrate how workplace conditions encompass norms and practices that can contribute to a limiting of worth, a prescribing of sacrifice, and an implying of the disposability of workers’ bodies. Ultimately, these two themes illustrate how occupational health and safety conditions are restricted by the meso and macro-level context of migrant agricultural workers. We believe that migrant agricultural workers’ workplace dynamics must be understood within a wider socio-political environment that influences their precarity and vulnerability. We end this paper by discussing the implications of our findings for programs and services that can better support the health and safety of migrant agricultural workers in British Columbia, particularly those arriving under the SAWP. 

### 1.1. Temporary Agricultural Programs and Workplace Vulnerabilities

The Canadian agricultural industry has a long history of depending on temporary foreign migrant labour. First started as a pilot program in Ontario 1966, the SAWP now hosts 74% of temporary foreign workers coming to Canada each year [1,6]. In British Columbia (BC), 11,468 migrant agricultural workers came under the SAWP, second only to Ontario’s sizeable 25,525 migrant agricultural worker population of 2018. Of increasing significance, another group of temporary migrant agricultural labourers are brought under the Temporary Foreign Worker Agricultural Stream, with 5362 entering British Columbia, and 18,095 entering Canada, in 2018 [6,12].

Workers under both of these programs face similar challenges that are well-documented in the literature and create risks to occupational health. Workplace challenges may be explicitly linked to health, such as a lack of adequate equipment or training on pesticide use, unsafe work conditions, strenuous/long work hours, and workplace harassment [10]. More implicit and complex workplace dynamics, and their influence on migrant agricultural workers’ health, have received less attention. These workplace dynamics and the wider context in which they occur, merit further attention because they significantly determine workers’ health, safety, and access to health services. Such occupational health threats include a limited ability to make use of legal protections, physical/sexual harassment or violence perpetrated by both employers and peers, and peer-policing and competition to perform at a rate that may cause acute or chronic injury [13,14].

Workers under the SAWP are re-hired based on an employer ‘naming’ system, whereby employers choose the workers by name they wish to hire in subsequent seasons. Such a system can create a coercive incentive for individuals to push themselves beyond their physical limits and to accept unsafe work conditions in order to secure a position the following year [15]. Lastly, workers’ sense of vulnerability, characterized by a fear of medical repatriation, social isolation and discrimination, inadequate access to health and other public services, and poor or inadequate housing can all impact day-to-day occupational health and safety [16].

### 1.2. Migrant Agricultural Worker Health in Context

MA workers face several challenges that threaten their physical and mental health. First, migrants are embedded in a larger context of global inequity that forces them to pursue work outside of their countries of birth. This larger context intersects with limited access to legal workplace protections and training, a work permit that is severely restricted, and a pressure to be ‘renamed’ (to return subsequent seasons under the same employer) that creates a climate in which workers may feel a ‘perverse incentive’ to accept poor working conditions, including hazardous or abusive environments [3,4]. Our research with MA women in the Okanagan for instance, highlighted experiences of heightened surveillance and workplace sexual harassment that typically was not reported to authorities [13].

Temporary citizenship status is also a significant barrier to MA workers’ health and wellbeing in the workplace. One study in BC for example, found that temporary migrants (in comparison to immigrants with permanent status) experienced more workplace coercion to take on exploitative or hazardous tasks by virtue of their limited mobility (e.g., living on farm, limited work permits) and the narrower range of labour protections available to them (e.g., no entitlement to overtime pay) [4]. Recent cases also indicate that MA workers’ attempts to assert their labour rights or pursue collective bargaining can be met with punitive measure that take advantage of their precarious migration and job status [5].

The fact that workers have little time or opportunities to build a social network off the farm can contribute to their vulnerability. MA workers are typically housed on their employer’s property (where they work), put in long hours, often experience language barriers, and face racial discrimination in their host communities [17,18,19,20]. Some research suggests that housing migrants on their employer’s property can create a disciplinary dynamic among workers that can promote competition or hyper-productivity that has a detrimental impact on their autonomy and dignity [14]. Migrant farmworkers may also be policed and segregated along racial, ethnic, and language lines, increasing isolation and exploitation [21]. Amidst these circumstances, access to transportation, and subsequently to health care or any other service, is challenging [10,13,22]. Further, migrants’ ‘foreign,’ or ‘temporary’ status exacerbates their segregation in the larger community, making it difficult to access, and even feel entitled to, resources and supports [13,17].

In addition to workers’ experience of isolation, scholars indicate that structural barriers manifest in their ability to access health services, the quality of care they receive, and in their overall health status. Weiler et al. [23] for instance found that workers had limited supports required to access healthy food, which consequently resulted in poor health outcomes for individuals in the South Okanagan. Hennebry et al. [22] found that in addition to long working hours and language differences, employer mediation could also have a detrimental impact on workers’ ability to access necessary services in Ontario. This has also been a barrier confirmed by both clinicians and migrant workers alike accessing services in the Okanagan Valley. Our interviews with clinicians in particular indicated that employer preferences often determined whether a clinician reported a workplace injury, and whether they protected the privacy of a worker’s health information. Literature in Ontario has also documented several instances of medical repatriation, cases in which workers were sent home because of a medical concern [24].

Several occupational health and safety challenges are uniquely experienced by migrant agricultural workers. Overcrowded housing, poor sanitation, and limited access to refrigeration for instance, all increase the risk of disease transmission [18]. Several agricultural activities including pesticide use and repetitive motions may expose workers to musculoskeletal injuries, eye and skin problems, respiratory conditions, sexual health concerns, and mental health challenges [25,26,27,28,29]. Furthermore, both short-term and cumulative exposure to animals [30,31], noise and vibration [32], airborne irritants [33], and stressful environments [34] can contribute to both physical and mental health consequences for migrant agricultural workers. Complicating the matter, there appears to be limited adherence to prevention and protective measures that could protect the occupational health of migrant agricultural workers on the job [33]. Attention to social and cultural norms, language and literacy needs, and employment-specific barriers may help address discrepancies in knowledge and behaviours of prevention measures in migrant agricultural workers’ workplaces [29,33]. In addition, the use of arts-based approaches, employer engagement, lay outreach/health workers, and community partnerships may hold potential to adopt a higher uptake of health and safety prevention strategies [35,36,37]. While these approaches and strategies are important, limited attention has been paid to the role that regulatory, economic, and political factors may play in shaping migrant agricultural workers’ sense of agency and ability to navigate workplace health and safety.

## 2. Materials and Methods

Our research employed a qualitative narrative approach, and first included 12 migrant agricultural worker participants from the Okanagan to better understand the experiences of this population. While we had initially planned to explore a different area of focus (belonging and wellbeing among diverse socio-cultural groups), workers’ accounts consistently emphasized working conditions. Consequently, we employed an emergent design to capture the importance of these experiences, and our key research questions for this area of inquiry were:What are migrant agricultural workers’ workplace experiences? And;What do these experiences tell us about workers’ health?

Using a narrative approach, we were most interested in workers’ perspectives of their workplace environment and the influence that they felt this could have on their overall health and access to health services. A narrative approach to research is focused on developing storylines or in-depth storied experiences that add meaning to our understanding of an issue [38,39]. In order to capture the richness of each participant’s experience, we invited individuals to participate in 2–4 one-on-one interviews and/or focus groups. Each interview took place in a community space that workers described as a place where they were most comfortable, and each interview lasted between 60 and 90 minutes. To help enrich the findings, we invited a key volunteer support person (*n* = 1) who had been working with migrant agricultural workers for over 10 years to participate in focus group discussions when she was available and incorporated additional one-on-one interviews with 11 migrant agricultural participants. In total we had 23 participants who were migrant agricultural workers under the SAWP. Of these, 18 were Mexican men, 3 were Mexican women, and 2 were Jamaican men.

While the first interviews were focused on following an open-ended line of inquiry that could capture the full range of experiences relevant to migrant agricultural workers, later interviews were focused on confirming key interpretations, themes, or storylines that had been expressed by these individuals. To establish a sense of reciprocity and trust among workers, a research team member volunteered in the region, helping workers with English classes and any issues that arose. These additional interactions with workers also helped build rapport and understanding among the team and participants, ensuring a more natural and open conversation style when engaging in interviews. This initial data collection occurred over a 2-year period from 2015 to 2017. Data was analyzed using a narrative framework first developed by McCormack in which participants’ accounts are reviewed several times through the use of multiple lenses: “active listening, narrative processes, language, context and moments” [40] (p. 219). As we reviewed data under each of these lenses, recurring ideas and stories were clustered together in order to organize these patterns into two cohesive themes. The sample size was characteristic of narrative inquiries (*n* = 24) which typically limit the number of participants in order to ensure the richness and in-depth exploration of each participant’s account.

In the following year during the peak of the agricultural cycle in the Okanagan Valley (June to October of 2018), we carried out consultations with 104 SAWP workers from Jamaica and Mexico in 3 different public locations for the purpose of confirming and contextualizing these findings, and to engage in dialogue with migrant agricultural workers to guide our future research direction. In addition, we conducted a content confirmation process with 8 key community volunteers actively working with migrant agricultural workers, to begin to understand the scope of key incidents reported by originally interviewed migrant agricultural worker participants. These follow-up activities further enhanced our confidence in the original narrative analysis. Here, we report on the key findings first developed through a narrative inquiry approach from 2015 to 2017, and further confirmed and elaborated upon via public consultations and content confirmation interviews with community volunteers in the summer and fall of 2018. All study activities received ethics approval from both authors’ institutional employers (UBC, harmonized review with Interior Health Certificate code: H18-02889; Okanagan College full board approval certificate: 19-005).

## 3. Results

In this study, we sought to better understand migrant agricultural workers’ everyday workplace experiences, and to examine how these experiences contributed to their overall health and safety. Our interviews with migrant agricultural workers revealed two key mechanisms that characterize everyday workplace experiences: (1) Authorities that silence, which encompassed two sub-themes: “it doesn’t make sense to say anything to them” and “no-one to enforce the rules” and; (2) “I will not leave my body here.” Each theme is presented in a narrative form below in present-tense, in order to capture a more emotive account of the experiences workers shared with us. A discussion will follow these narratives.

### 3.1. Authorities That Silence

“It doesn’t make sense to say anything to them.” The Seasonal Agricultural Worker Program gives the sending country government jurisdiction over the workers in their programs even while they are in Canada [5,7]. Migrants under the SAWP are also entitled to most workplace protections under Canadian law, in addition to human rights and housing protections. Therefore, both sending country officials and Canadian bodies have the authority and responsibility to protect these individuals. Workers are instructed that complaints or concerns about working conditions need to go through their consulates. Yet workers’ accounts reveal that the consular officials’ loyalty is often divided, if not completely lacking. They share stories of their consulate acting as gatekeepers, dismissing them, or failing to be effective at protecting their rights. Reflecting on their working conditions, Javier reflects, “It’s the consulate who should be monitoring ... but they don’t do it.” He adds: “The reality is that the consulate, they already have these reports, of how our bosses treat us, and they haven’t done anything”.

Elena had not had much reason to call on the consulate although many of her peers had complained about them. “But then when I did need them, I knew all the other Mexican workers were right to complain. The consulate doesn’t do anything.” “Because you experienced it first hand?” a research team member asks. “Yes, first hand. And that is how I realized that it’s all a lie. They don’t help you at all,” Elena says.

Some workers conclude that the program only brings more restrictions and fails to deliver on the protections that it promises. Antonio says, “supposedly we are protected by this program, but we, [are] better even if illegal [in the United States]”. Other Mexicans we meet who have worked in the United States agree, some feel that there are more established networks of support there with the influence to address labour concerns. Or at the very least, if they do not like their treatment by one boss, they can try to find work on a different farm or workplace. Feeling that they lack a formal advocate in Canada, workers feel coerced to accept their situation. “At the moment, we don’t have another option,” says Pablo.

Many have lost confidence that their government representatives will look after their best interests. Wayne recalls a time when the Jamaican liaison came to visit the farm where he was working. “She come here one time, with the boss and his wife ... just talking through the fence ... they were on the road we were in the field.” This dynamic was commonly reported by Jamaican workers. The liaison is seen as an enforcer of the employer’s dominance, insisting that workers conform and dismissing their concerns. Wayne continues “... most times they [Jamaican liaisons] don’t know. Most times it don’t make sense to say anything to them. Because most of the time, [they say] ‘ooh you are [a] big man, you can deal with it’. You try to deal with it they say, ‘’ooh, guys there are 3500 guys in Jamaica sitting here waiting for your position’”. Wayne continues, “If you try to speak and stand up for your right, I’m going to pay tough for this.” The message seems to be, you take what you can get and you don’t complain, and if you do, you are not coming back.

Therefore for participants, control and dominance is palpable in many interactions with the employer and consulate alike. In part, the boss’ power comes from the workers’ need to keep their job at almost any cost. In some cases, even when workers have expressed to us that they have been treated poorly, many of them want to be “renamed,” to be identified by their employer as a preferred employee to return. Siriaco shares that many of them “are with that fear, that things could be worse ... [and] we fear complaining to them, to say [anything], [because] ... they may say, this guy is too combative, we will not ask for him [back again].”

“There is no-one to enforce the rules.” A research team member is asking workers about what strategies could improve conditions for them. One worker contemplates whether the boss understands the obligations that he has to his employees. Lina, a long-time volunteer in the area intervenes:
“*I think [the problem is] because there is nobody to enforce the rules. Look, there are rules and regulations for them [the workers], there are rules and regulations for the bosses, but there is no one, not one inspector that comes all of a sudden and says, ‘look, Ms. Smith brought 4 people [workers] from the same farm to the hospital with the same symptoms. And none of these men have masks, but the boss has a mask and glasses [protective eye gear] when he is spraying [pesticides].’ It’s against the law that these young men not have this when they are working … I’ve seen it, but who am I? So then, he [the boss] says to me, ‘no no, there are masks, it’s just that the men don’t want to use them’. And they [the workers] say to me, ‘when have they given us masks if there are none here?’ So there is [virtually] no inspector. And the bosses know [that]*.”

She tells us about another time that she translated for an inspector. At the time, she asked the workers questions:
“‘Guys, do you have masks?’ They were dying laughing ... ‘You have tables where you can sit to eat?’ ... I tell Samuel [the boss], they don’t have any of this. ‘Yes, yes, yes, but we are going to get it.’ ... [he responded]. Where is the sanitizer for their hands?! These inspectors must either have agreements [with the farmers] ... because the only thing they do is sign papers, and then they leave for another year.”

Workers at the table have their own stories about the limited oversight that occurs (i.e., consulate and inspector visits). Like Lina’s experiences, inspections are always announced to the employers beforehand. José recalls a time that his boss knew that an inspector was making a visit. There was a day of preparation for the visit, more stoves than what had actually been provided to workers were brought in, and pesticides, chemicals, and debris were removed from the living quarters in order to appear to meet housing requirements. At another farm, a worker shares:
“Well, I’ve seen this in my boss’ farm, well, in particular, with the inspecting of the cherries right? But they tell him a date that the inspector is coming, and then, they know and they start preparing. Actually, they even told us,‘So, you know this day, you’re going to have to use gloves’.”

Other workers laugh, interrupting, as they recall their own experiences of being expected to perform for the inspector. “What is needed is that they show up unannounced,” says Estefanio. All the workers around the table express agreement.

In a location north of Estefanio’s farm, Jamal draws the same conclusion:
“We need liaison officers to pay more attention to us. Don’t call the boss when you’re coming. Just turn up on him. Surprise him. … You come today. Ok ... I’ll be in that same spot tomorrow again ... we sign our contract and most of the things, and most of us, we’re all breaking the contract. If we are going to sign a piece of paper, we all should stand by it. Most of us doesn’t get time to read it until we are here ...”

Like most workers we interview, Jamal wants to see more involvement from his consulate and he wants random inspections so that employers do not have a chance to cover up their failure to keep their workplace obligations. The worry for Jamal is not just for his physical safety in the workplace, but also for the complicity that his employer forces upon him when he asks him to carry out tasks or activities that he is not permitted to do under his contract, and as a result, that he is not trained to perform.

In a follow-up interview on José’s farm, we meet with the same group of workers. We want to discuss what needs to change in regards to workplace safety. “Here, they haven’t touched us,” Vicente says, referring to the consulate. “The consulate should be in charge of all of that [safety]. So far, I don’t think they have done a very good job in this zone...” They wonder how the consulate’s presence might be different in other regions of the country. Jorge shares a story about a friend located in eastern Canada. His friend had not been given “a dignified living situation and the young man ... when he insisted to the boss [that living conditions be improved], the boss said, ‘I will not ask you back.’ In that sense, the consulate doesn’t work. Because he had asked the consulate to come and see how he lives [and they didn’t come],” explains Jorge. Stories like these are shared often, some of them much closer to where the workers reside now. They send a message to workers: there are rules that must be followed, and then there are rules that protect the worker. In many cases, a worker’s rights are dependent on what their boss can get away with.

### 3.2. I Will Not Leave My Body Here

Juan states, “without the worker there is no fruit, and without the employer, there is no work.” The statement is a truth echoed from many mouths. Yet does the employer honour this interdependence or will any body do? “I will not leave my body here and with the more and more he wants ... this is where it ends,” Ernesto declares. “He already fired two. And one he hit in the face with apples. And the consulate did nothing for him,” shares Mario, a worker from a different farm. He tells us about a worker, a good worker, he says, who the boss had no patience for. It was his first time working in the temporary migrant agricultural worker program. Mario advocated for him, telling his boss, “give him a chance,” to find a rhythm, to learn these new tasks. The boss would deny him work when he was not satisfied with his performance. “He managed a week without work ...” Mario says. The worker did not stay.

Ultimately, the workers have a difficult relationship with their body. Their bodies are sacrificed in exchange for a job that wants them only for as long as their bodies are intact. They need the work, but they need their bodies to do the work, and often, they are asked to push themselves to their limit, work through pain, hunger, injuries, endanger themselves on old ladders, ride on the back of trucks, work without breaks, work without safety gear or adequate protection, work 14–16 hours daily, often, without days off. On one farm, workers tell us about unsafe and unhygienic practices that they are expected to do. Rodrigo says “on top of the ladders, we are endangering ourselves ... it’s a huge mistake... we’re going to kill ourselves [but] that’s where they put us.” Salomon, from a different farm, questions them, “but why do you do it?” No answer is given. Perhaps the question was meant rhetorically. Either way, the silence is understood.

Most workers express concern that their boss is always ordering them to go faster and they fear that their body will be permanently damaged as a result. Gerónimo reflects:
“I am quite fast ... I have done this for many years ... but what do I gain? I will leave my body here.” Andre shares, “the hardest type of work is here [in agriculture], go go go go go go go go yeah ... because there is someone behind you going go go go go go go go go. But oh, how fast I am going to go. ... or how long am I going to go, after 9 h, 10 h, if I go go go go? Oops, at the end of the day, I’ll end up on a plane going home. I’m going too fast. I’m not a machine. I’m not a machine... If you have a blender in your house and you just press it and someone just leave it there, it has no use. How long is it gonna run at that high speed blending whatever ice, juice, how long is it gonna run, ‘’eeeeeh,’ it’s going to burn out. Man can replace those things. But I can’t be replaced. I can’t get a next heart tomorrow morning. I can’t get a next hand that’s going to work just like this one.”

Notably, workers talk about the unfairness implicit in receiving an hourly wage as opposed to the piecework rate offered to Canadian workers. Workers are entitled to whichever rate of pay gives them a greater return. Yet these individuals often report being evaluated as if they were being reimbursed by the yield of what they pick (e.g., per box of apples or cherries) but only compensated per hour of work (i.e., minimum wage). This creates an exploitative pace of labour in which workers may endanger their health and safety more for the fear of losing their job rather than because of a positive economic incentive. Further, none of these workers are entitled to overtime yet as long as their employer can push them faster, they will get more ‘value’ for the time that a worker spends on the job.

## 4. Discussion

Our findings identified two over-arching themes that largely determined workers’ occupational health and safety environments. In Theme 1, authorities that enable and silence, migrant workers described ways in which authorities intended to protect them, particularly the consular officials and employers, either actively or passively helped reinforce coercive power dynamics, and/or, failed to deliver on protections promised under the Seasonal Agricultural Workers Program in Canada. The subtheme of “it doesn’t make sense to say anything to them,” revealed workers’ sense of powerlessness in the workplace. Of significance, accounts revealed that authorities enforced workers’ conformity rather than empowering them to assert their rights to adequate health and labour standards. The subtheme of “there is no-one to enforce the rules,” illustrated the limited visibility and presence of supports and authorities ensuring adequate health and safety standards in the workplace. In many cases, workers reported not having any contact with authorities or inspectors who would be in a position to monitor or regulate workplace conditions. Participants also distinguished between unannounced and announced inspections, articulating that the norm of announced inspections in the workplace created a permissive climate in which poorer working conditions could go unchecked. Without regular monitoring of their work environments, the onus was on workers to self-report workplace hazards that were putting their health at risk. Yet individuals typically did not feel empowered to report these concerns, and in the cases that they did report them to authorities, they were often made to feel more vulnerable than protected.

Prior research has identified various layers of vulnerability that contribute to a coercive and unhealthy work environment [15,16,21]. Such research requires us to consider conditions such as precarity (e.g., in terms of dependence on another for permission to stay in Canada), freedom (e.g., a prescribed place of residence) and segregation (including familial separation and geographic isolation) as key determinants of health and healthcare access for migrant agricultural workers. Considering these conditions in light of our current findings, it is clear that stories shared under the theme of “authorities that silence” are so prominent in workers’ lives precisely because alternative pathways and supports towards health are so clearly restricted or limited. In short, without the freedom, power, or opportunity to speak up about health issues in the workplace or to refuse unsafe work or poor living conditions, then the project of improving this population’s health must be closely tied to the politics of stabilizing security and status for this group.

These vulnerabilities are also felt simultaneously at regional, national, and international levels. For example, the local economies of Caribbean and Mexican countries have been largely devastated by liberalized trade agreements and mega-projects. As these local economies are undermined, workers of the Global South have been displaced or relocated in order to find economic means to sustain their families, and a more captive, and even desperate, workforce is created [10,40,41]. Yet specifics of the program also play an important role in creating coercive and under-monitored workplace conditions. The practice of being tied to a specific employer (being ‘renamed’) for instance, has been criticized because it places unique power in the hands of the employer to determine not just a worker’s ability to stay employed at a particular work site, but by extension, their status in the program, and ability to remain in Canada [16]. Advocates have suggested policy alternatives such as open work permits and the granting of permanent residence upon arrival [42]. Both of these policies would enable workers to increase their workplace mobility, and consequently, better empower them to refuse unsafe or hazardous work practices that threaten their most basic physical and mental health. If workers are able to refuse unsafe work, perhaps by extension, there will be increased incentive for employers to improve their workplace environments. Yet the option of permanent status upon arrival would also allow workers the opportunity for greater protection under the law, and would remove jurisdictional blurriness that maintains workers at the margins of legal protections [43].

Scholars have long criticized the inadequacies of complaint-driven mechanisms that take a reactive rather than a proactive stance to the regulation of workplace conditions [44,45]. Under pressure from advocates and migrant rights groups, the Canadian government has committed to increasing unannounced inspections to provide greater oversight of substandard workplace and housing conditions faced by migrant agricultural workers [1]. These regulatory practices would have some potential to strengthen oversight of the program and even increase workers’ confidence that they can assert their rights on the job. Yet as noted in the accounts above, it would require adequate staffing, resources, and political will to implement such approaches, both by Canadian authorities and sending country consulates. Complicating the calls to improve the oversight and regulation of the program is that workers who have experienced a great deal of hardship may carry a mentality of “better the devil you know.” In fact, our consultations confirmed that most workers had experienced similar challenges, but that they were reticent to leave their bosses for fear that their next workplace could be even more difficult or unsafe. Ultimately, this suggests that even if ideal regulatory mechanisms intended to protect migrant agricultural workers are established, it will require that officials first regain the trust and confidence of workers who have been failed by regulatory authorities in the past. This theme also suggests that if workers’ access to protections continues to be limited and perceived as both systematic and routine, these individuals are likely to normalize risk in the workplace, further endangering both their physical and mental health.

In Theme 2, “I will not leave my body here,” participants’ stories revealed ways in which their economic and political status created lose–lose conditions in which they felt compelled to sacrifice their physical health for short-term security in the SAWP while at the same time endangering their long-term livelihood in this same program, and even their basic survival. Workers expressed a deep ambivalence about this relationship, both expressing acts of defiance or intentions to refuse unsafe work conditions, while still not feeling at liberty to refuse unsafe conditions or labour standards. Cohen and Hjalmarson’s prior research indicates that even amidst coercive labour conditions, migrant agricultural workers are active in creatively resisting exploitative conditions and that they are not merely passive participants in agricultural programs [46]. Nonetheless, the material consequences of living within this coercive climate have inevitable consequences for the health and safety of migrant agricultural workers. Participants in this study were reluctant to report workplace injuries, refuse unsafe workplace practices, seek medical attention for workplace injuries, and assert their rights to adequate hygiene (e.g., handwashing stations, washrooms) and prevention measures (e.g., seatbelts, breaks, protective equipment). A lack of oversight and will to ensure that these measures are in place in the workplace will continue to have significant consequences on the physical and mental health of migrant agricultural workers.

Given the health risks and challenges faced by this population, workers’ ability to access health care services also merits attention. Participants in our study described various barriers to accessing medical care: transportation, geographic isolation, language, limited clinic hours (in combination with demanding work hours), and fear of job loss or medical repatriation. In Ontario, a survey of 6000 migrant agricultural workers indicated that many workers fear, or actually suffer negative consequences because of seeking medical attention. In fact, 44% of respondents stated that a co-worker would work while sick for fear of telling their employer, and one in five reported that their boss had expressed anger as a result of their illness [22]. While no large surveys have been conducted in BC, the qualitative accounts identified in our narrative inquiry and confirmed in our subsequent consultations, suggest that a similar dynamic can explain the limited healthcare seeking in the Okanagan Valley. Beyond replicating accounts that have been documented in different regions, this theme helps uncover the true motivation behind workers’ limited health-seeking behaviours. This theme challenges commonly held views that workers’ cultures, lack of trust in the Canadian system, or lack of information prevent them from seeking healthcare or related protections [47,48]. Rather, workers navigate a coercive environment in which they feel that they must choose between their health and their livelihood.

Even in cases when individuals do access medical care, they encounter various challenges that may negatively impact the quality of care they receive. Our interviews with community volunteers accompanying migrant agricultural workers to hospitals and clinics helped to further illustrate these barriers. Volunteers and migrant agricultural workers alike reported a lack of access to an unbiased, third-party translator, for instance, making it difficult for workers to fully disclose their symptoms, especially if the only translator available held a supervisory position in their workplace. In these cases, workers feared disclosing the severity of their symptoms or reporting workplace injuries because it could affect their job status or put them at risk for medical repatriation. In other cases, interviews and content confirmation carried out in our research indicated that employer translators may have a vested interest in downplaying symptoms and preventing the reporting of workplace injury claims that may have financial or regulatory consequences for them. Failure to provide independent care not mediated by an employer may exacerbate vulnerabilities or limit the protection workers are afforded. Fearing punitive consequences, workers may hide workplace injuries, accept unsafe work practices, and not seek necessary medical treatment. We have even witnessed workers participate in their own medical repatriation, not because it is their preference, but rather, because employers feel threatened by the possibility of a worker seeking compensation for a medical injury. Mirroring our findings, accounts of employer mediation have also played a significant role in limiting health care access and delivery to migrant agricultural workers in Ontario [22]. In addition to anticipating barriers to access and quality care, our theme of “I will not leave my body here” suggests that a more complex approach to healthcare planning that can account for the deep ambivalence that workers may feel about care provision must be developed. For instance, changing standard practice for this group to more proactively screen for workplace injuries, preference for the place where care is provided (e.g., Mexico or Canada), and an explicit reminder of what workers are entitled to, could help foster greater cooperation among migrant workers and their host-country care providers.

Many participants reported out-of-pocket expenses and difficulties receiving compensation by their private insurance company when direct billing is not available in clinics that they access. In British Columbia, the premium rates that residents pay to access provincial health care coverage make access to public health services cost-prohibitive to migrant agricultural workers, even when/if they meet the eligibility requirements for public health coverage [13]. Notably, as of 2020, these monthly premiums will be eliminated, which may have the potential to begin to address healthcare coverage gaps faced by migrant agricultural workers [49]. However, it is important to note that even with fewer financial barriers to registering for provincial health programs, other barriers remain that mean many workers still may not have access to such coverage. For workers in Ontario, notable challenges include the three-month waiting period and their dependence on their employers to assist them in registering for provincial health coverage [22]. Such developments and challenges remind us that healthcare barriers and pressures faced by workers under the theme of “I will not leave my body here,” are very much woven into the equity and governance issues that were revealed in stories of “authorities that silence.” Ultimately then, short and incremental steps towards addressing gaps in healthcare prevention and care provision for this population are warranted, but without attention to the larger social and economic factors that shape and undermine migrant agricultural workers’ living and working conditions in Canada, this group will continue to face significant health and safety challenges.

## 5. Limitations and Considerations

As we write this paper, important changes to labour permits for migrant agricultural workers are currently underway [7,50] and should be considered when interpreting these results. Furthermore, given that the focus of the analysis was on workers’ stories, this study is unable to provide a rich account of service providers’ and authorities’ perspectives of their support roles. A study of such accounts may help provide further understanding of socio-political factors that shape the health and safety of migrant agricultural workers.

## 6. Conclusions

The findings presented here illustrate the ways in which migrant agricultural workers’ workplace environments are significantly mediated by political and economic structures, such as globalizing forces that have undermined local economies in the Global South, and the limited scope, and in some cases, complicity of sending country officials that fail to protect workers’ basic rights under the program. However, these macro-level structures do much more than determine transnational oversight mechanisms. They are also reflected at the level of the farm and the worker under these programs, creating coercive, paradoxical, lose–lose choices in which workers may normalize, accept, or only partially resist unsafe or hazardous workplace environments. Provincial and regional bodies that do not interrupt these conditions of coercion, may wittingly or unwittingly reinforce these dynamics, by, for example, relying on complaint-driven mechanisms that place the onus on workers to take on further risk in the attempts to address inappropriate workplace practices. Likewise, healthcare and other public sectors may create further conditions of coercion by posing barriers for workers to access or navigate services, which may require them to confront further risk to their job status, or, long-term health. Ultimately, workers’ right to a healthy, hazard-free workplace must account for the various political, economic, and regulatory factors that currently limit and undermine this very possibility.

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
