# Peer review of "“I Will Not Leave My Body Here”: Migrant Farmworkers’ Health and Safety Amidst a Climate of Coercion"

_ijerph, 2019, doi:10.3390/ijerph16152643_

Round 1

Reviewer 1 Report

The authors present a compelling analysis of the structural vulnerability of temporary agricultural workers in British Columbia.  Their analysis adds to the growing literature that demands major changes in the way that agribusiness and government regulatory offices treat agricultural workers across North America.  Although the authors offer the core of an excellent paper, addressing three major concerns could improve their presentation.

Materials and Methods.  The authors discuss how they collected data.  The authors need to add how they managed and analyzed these data to their discussion.  They also should provide some information about the agricultural workers who participated in their study.  This information should indicate the total number of workers, their genders, and their national origins.

Discussion:  The authors integrate their results into the literature.  At the same time, their discussion of access to health care (p9, line 417 – p10, line 456) goes well beyond the findings they present in Results.  I do not disagree with their presentation, but they present new results in this discussion that should be presented in Results.

Limitations:  The authors do not provide a discussion of their study’s limitations.

Author Response

Thank you for your thoughtful feedback. We feel very strongly that making changes to our manuscript based on our feedback has strengthened our manuscript. Please see detailed responses and a list of changes in the attached document.

Reviewer 2 Report

Over all, I really enjoyed this paper. I do think that there is some work to be done regarding the clarity of the argument, which got a little bit lost. This was an outcome of writing-style, but also the structure of the paper, which was a little circular. I've done my best to provide the authors with more detailed comments and suggestions below. 

Regarding the writing--on the whole, the paper was comprehensive and coherent. I would suggest small changes: "our research focuses on" (line 34) as opposed to "our research was focused on". Or, "we examine how workplace environments impact on agricultural workers'  health" (line 36-37) as opposed to "we sought to examine how..." -- things of that nature that will streamline the text. (Also, while in the introduction the authors use the past tense, in the findings, they use the present tense; I wonder if this could be made consistent.) 

Similarly, line 56-57 is a bit cumbersome with "accessing" and "access" being used in the same sentence. 

Line 71: How do the stories mitigate migrant workers' ability to ensure a safe and health workplace? This phrasing is confusing. It is the coercion that mitigated that ability, not the stories. 

While some sections are well cited, others require additional referencing. For example, line 220-223. 

Content 

What precisely is meant by "health"--does it include both physical and mental health? Line 114, "well-being" is added? Are health and well-being interchangeable? 

I really liked the "I will not leave my body here" theme. It was really compelling. I did, though, find the authors' introduction of the themes to be a bit clunky and difficult to follow. Modifying the text would likely help in this regard. So, for example, the first theme "authorities that enable and silence" is not easily understood. "Authorities that silence", maybe? But the enabling could be positive or negative, so in this context and read through the sub-themes, use of "enable" is a bit confusing. 

Also, lines 74-76, I wonder if a line or two specifying how these things are illustrated might not help. As it is, the reader is asked to simply accept these statements without any sense of why. 

Line 78-80, I wonder about the use of "contextualize" here--is that what the themes are doing? Maybe more accurately, the themes reveal something about the conditions that limit the ability to MA workers to access their rights, etc.? 

Line 44-45: Could you specify who these third parties are and perhaps alter this description of the TFWP for clarity. Also, line 46-47--are the unique implications and common challenges that exist across these two programs relevant to the argument you're advancing? And perhaps, either way, they should be stated here. As it currently reads, I anticipated that discussion in the next paragraph. 

Having flagged these two streams, it becomes a little confusing when you move into the discussion of numbers of approved positions (line 48-51), etc. Are these SAWP positions? TFWP Ag positions? Or both? Given the composition re country of origin, I suspect SAWP, but this needs to be clarified. Later in the paper  (line 93), the authors flag the distinct features of these programs again, but in what follows, only offer one distinct feature (that had been previously mentioned). Again, if these distinctions are important, I think they should be more fully developed. 

Line 102-105 seems really important regard the paper's contribution. I wonder if this could be forefronted. In a way, this IS the work of the paper: figuring out the workplace dynamics that impact MA workers' health outcomes. 

Line 106-107--this "additionally" is confusing given that the preceded paragraph started with a statement about the common challenges faces both groups of MA workers. 

Section 1.2 is very nicely done.

Section 3.1 Authorities that enable silence--I feel like this section would be strengthened with some additional framing. So, you're talking about two sets of authorities? Maybe more, and flagging that at the onset would make the section more robust. 

Line 224-225, I wonder if there isn't a distinction between "dismissing" the workers outright and "failing to protect them"--so I suspect that sometimes these things happen concurrently, but are there times when the authorities in the country of origin are simply unable to respond (due to their own structural constraints, etc.) to complaints? So, thinking here about the (global) hierarchies (flagged earlier in the paper) that underpin the mobility of these workers in the first place. 

Line 284--what is meant by "extra stoves"? 

Section 3.2--I think this is a really interesting contribution, but it's inadequately developed. And can the link between this insight and wages be elaborated (line 350-353). 

Section 4 The discussion is under theorized. The authors draw on a number of sources to support their claims, but these are not elaborated. 

Moreover, and perhaps more importantly, the connection to health is underdeveloped, such that first four paragraphs of the section don't discuss health or the link between the workers' inability to access protection and their health outcomes. This is remedied beginning on line 403 as the authors begin to unpack the second theme, but even here, the discussion duplicates much of what is offered in the introductory sections of the paper. And again, these is a lack of theorizing and integration of the scholarship. Finally, the ability (or rather, inability) of workers to access health care is included in the discussion and it's also flagged earlier. But it's not clear how it connects to the themes that are, otherwise, prioritized by the authors. I think some additional attention to the analysis/discussion needs be paid before the paper is accepted for publication. 

Author Response

Thank you very much for your thoughtful and detailed review of our manuscript. We were happy to have the opportunity to address your feedback and ultimately, develop a stronger manuscript! Please see a detailed list of changes and responses attached.

Round 2

Reviewer 1 Report

The authors have responded to all earlier comments.  No further comments.

Author Response

Hi there,

I know you had no further comments but I am attaching my responses to reviewer #2. Thanks for your time!

Reviewer 2 Report

The authors have done a good job tending to the earlier suggestions. The paper is much more cohesive and coherent, and the analysis as it pertains to health is much clearer. A few last comments--the most significant of which focus on LINE 77, LINE 180, and LINE 509. For each, I don't suspect much work is required. 

Line 18 The abstract needs to be changed to reflect the change to theme 1 (Line 71 + Line 220) (the "enable" remains in the abstract but is absent from the discussion of the themes in the body of the paper). 

Line 77 I'm still unclear on how the themes help to "illustrate how workplace conditions position workers to either accept, negotiate and/or resist exploitative norms or unsafe practices". I think the connection between this acceptance, negotiation, and resistance to the two themes needs to be more fully (if briefly in the context of the introduction) developed. Also, aren't those norms and practices underpinning the conditions? So, thinking here about how the sentence is structured and a bit circular. 

Line 100 This is a good addition. 

Line 180 I think there needs to be a greater elaboration of how these questions resulted in the data and subsequent analysis that you offer in this paper. These questions, it seems correspond to a larger project and not just the themes / analysis offered in the paper. I simply raise this because this paper does not, it seems, deal with health outcomes as such, but rather on workers' understandings of those systems that impact on their health (theme 1) and their subjective understandings of their own positions within those systems (theme 2). 

Above I've flagged that the methods as described need improvement. Here, I don't mean altered, but rather that the methodology section should be modified to reflect the findings of this particular paper. So, adding a line here or there that signals how you reached these particular findings in the context of the broader project. 

Line 509 (limitations)--could this be framed differently. It's very self-deprecating and detracts from the otherwise strong quality of the work. Moreover, it reinforces qualitative norms and appraisals of what constitutes sound research. I wonder if articulating the strengths of a small sample, of the methodology, etc. might not better serve the paper. Also, perhaps it should come under methodology. As it is, you have the reader convinced of what you're saying, but then, AH! at the end you say "oh but maybe it's actually not that convincing" :) 

All the best! 

Author Response

Please find form attached. Thanks!
